# Evaluation of Individuals with Non-Syndromic Global Developmental Delay and Intellectual Disability

**DOI:** 10.3390/children10030414

**Published:** 2023-02-21

**Authors:** Rowim AlMutiri, Maisa Malta, Michael I. Shevell, Myriam Srour

**Affiliations:** 1Division of Pediatric Neurology, Department of Pediatrics, McGill University, Montreal, QC H4A 3J1, Canada; 2National Neuroscience Institute, King Fahad Medical City, Riyadh 12231, Saudi Arabia; 3Division of Child Neurology, Department of Neurology and Neurosurgery, Federal University of São Paulo, São Paulo 04024-002, Brazil; 4Research Institute of the McGill University Health Center, Montreal, QC H4A 3J1, Canada

**Keywords:** global developmental delay, intellectual disability, genetic testing, non-syndromic intellectual disability, clinical evaluation

## Abstract

Global Developmental Delay (GDD) and Intellectual Disability (ID) are two of the most common presentations encountered by physicians taking care of children. GDD/ID is classified into non-syndromic GDD/ID, where GDD/ID is the sole evident clinical feature, or syndromic GDD/ID, where there are additional clinical features or co-morbidities present. Careful evaluation of children with GDD and ID, starting with detailed history followed by a thorough examination, remain the cornerstone for etiologic diagnosis. However, when initial history and examination fail to identify a probable underlying etiology, further genetic testing is warranted. In recent years, genetic testing has been shown to be the single most important diagnostic modality for clinicians evaluating children with non-syndromic GDD/ID. In this review, we discuss different genetic testing currently available, review common underlying copy-number variants and molecular pathways, explore the recent evidence and recommendations for genetic evaluation and discuss an approach to the diagnosis and management of children with non-syndromic GDD and ID.

## 1. Introduction

Global developmental delay (GDD) and intellectual disability (ID) are frequent referrals for clinical evaluation by pediatricians, neurologists, and geneticists. GDD is defined as a delay of two standard deviations below the mean in two or more developmental domains (e.g., motor [gross/fine], speech/language [expressive, receptive, mixed], cognition, personal–social, activities of daily living) [1] and is used in reference to children below the age of 5 years. ID refers to deficits in intellectual (such as in reasoning, verbal comprehension, abstract thought, comprehending instructions and rules, memory, problem-solving and learning from experience) or adaptive functioning evident in early childhood [1]. Neurodevelopmental disorders (NDD) are a group of brain-based conditions that first manifest in early childhood, that are characterized by evident impairments or limitations in personal, social, academic or occupational functioning; that encompass GDD and ID; and that include other entities such as autism, cerebral palsy, developmental language impairment, developmental coordination disorder, learning disability and attention deficit and hyperactivity disorder [1].

In the United States and Canada, 3–5% of children meet the criteria for GDD, and 2–3% eventually fulfill the criteria of ID when they are more than 7 years of age and when standardized intelligent quotient (IQ) testing can reliably diagnose ID [2]. The prevalence of ID is variable across different global populations, spectrum of severity and socioeconomic levels [3,4,5]. In a recent study, the global prevalence of ID was found to have decreased from 1.74% in 1990 to 1.39% in 2019 [5]. Given the overall increase in the global population, the total number of individuals with ID was larger overtime [5]. There is a higher proportion of individuals with ID in the regions of low–middle socio-demographic index compared with high socio-demographic index (five–six times higher) [5]. The higher prevalence of ID in developing countries and low socioeconomic areas was postulated to be due to higher exposure to risk factors and environmental influences [5].

Many children first diagnosed with GDD will later be diagnosed with ID, and conversely, many children with ID were first diagnosed with GDD. Approximately two-thirds of individuals have mild to moderate ID, and one-third have severe to profound ID. GDD/ID is often classified as either syndromic or non-syndromic (or isolated), whereby patients with syndromic GDD/ID present with additional clinical features (e.g., dysmorphic features, epilepsy, ataxia or focal neurological/motor deficit), malformations or co-morbidities (e.g., congenital malformations, growth abnormalities or involvement of other systems), including metabolic manifestations, while patients with non-syndromic GDD/ID have GDD/ID as the sole clinical feature [3,6]. The distinction between syndromic and non-syndromic GDD/ID can be relatively arbitrary, as additional manifestations may be subtle or overlooked, or may evolve and become more evident over time.

The etiology of GDD/ID is very heterogenous and encompasses both genetic and acquired causes. Hypoxia, infections and exposure to various environmental toxins (e.g., alcohol, illicit drugs and heavy metals) during pregnancy, in the perinatal or postnatal period underly between 11 and 55% of GDD/ID [7,8,9,10,11]. Genetic causes, identified in up to 47% [10,11], include chromosomal abnormalities/rearrangements (e.g., aneuploidies), small copy number variants (CNV) and monogenic causes with more than 1000 involved genes thus far identified [12,13,14]. All types of inheritance patterns (autosomal dominant, autosomal recessive and X-linked) are associated with GDD/ID. In a large cohort of individuals from an outbred population with unexplained NDDs, where GDD/ID constituted the most common phenotype, de novo pathogenic variants (i.e., absent in the parents) were found in 42–48% of affected individuals [15], and autosomal recessive causes were responsible for only 3.6% [15]. In contrast, autosomal recessive causes underlie larger proportions of GDD/ID in populations where consanguinity is common, estimated at 51 to 80% [16,17,18,19].

A specific and timely molecular diagnosis in GDD/ID is of great value to patients, families and treating physicians, as it allows a definite diagnosis for the patient, an end of the often-prolonged diagnostic odyssey, improved prognostication, as well as more accurate genetic counseling, estimation of recurrence risk and prenatal/preimplantation genetic diagnosis [20]. In addition, a specific diagnosis can result in improved care and management, with a modification of targeted surveillance efforts and an avoidance of unnecessary investigations. Furthermore, it may improve a patient’s and family’s access to community services and the evolving network of rare diseases support groups [21].

Guidelines for the genetic evaluation of individuals with GDD/ID are similar, though not homogenous. Most of the recommendations were developed prior to the wide-spread clinical availability of next-generation sequencing-based tests such as exome sequencing and comprehensive gene panels. Almost all professional organizations offering guidelines (American College of Medical Genetics, American Academy of Neurology, Child Neurology Society, American Academy of Pediatrics, American Academy of Child and Adolescent Psychiatry) have consistently recommended chromosomal microarray (CMA) and Fragile X testing in as a first-tier test for children with GDD/ID, especially when no specific syndrome or disorder can be readily delineated [22,23,24,25]. However, there are wide inconsistencies with recommendations related to biochemical screening for inherited metabolic disorders (IMDs) associated with GDD/ID [10,24,26,27].

In this review, we will discuss the available genetic tests, their yields as well as the common molecular pathways underlying GDD/ID disorders. We will also outline an approach to genetic testing and management in children with non-syndromic GDD/ID.

## 2. Genetic and Metabolic Investigations in Children with GDD/ID

### 2.1. Chromosomal Microarray

Chromosomal microarray (CMA) uses comparative genomic hybridization or single nucleotide polymorphism (SNP) testing to detect chromosomal copy number variants, i.e., gains and losses of chromosomal material. CMA can detect CNVs as small as 20–50 kb and has almost completely replaced the use of karyotyping. However, CMA cannot detect balanced chromosomal rearrangements and has a limited ability to detect low level mosaicism [28,29,30]. CNVs of uncertain established significance can be difficult to interpret. Furthermore, CMA may also detect incidental findings, unrelated to the primary indication. Finally, there are also individual susceptibility CNVs, with documented variable expressivity and/or incomplete penetrance, which are also frequently inherited [31,32,33]. The clinical utility of these susceptibility CNVs is not yet well-established [32].

The diagnostic yield of CMA in NDD overall, including GDD/ID, ranges between 10 and 20% [34,35,36,37]. The diagnostic yield of CMA is lower in individuals with mild vs. moderate to severe ID, with reported yields of 12–19% vs. 20–30% [38,39,40]. A few studies have explored CMA diagnostic yield specifically in non-syndromic GDD/ID, with a reported yield of 10.9% [41]. Note that karyotype analysis is needed to detect balanced translocations, with a diagnostic yield of 3% in cases of developmental disabilities of unknown cause [42]. All current society guidelines recommend CMA testing as a first line [22,23,24,25].

### 2.2. Exome Sequencing (ES) and Comprehensive GDD/ID Gene Panels

Exome sequencing is a massively parallel gene sequencing approach, also termed next generation sequencing (NGS), that allows examination of the DNA sequences of most of the protein-encoding exons (~1.5–2% of the genome) of an individual. ES has a few important limitations. Coverage of the exome is not complete and may vary between laboratories and technologies; therefore, not all exons are examined, potentially affecting testing yield. In addition, ES does not reliably detect mosaic variants, exon-level deletions, repetitive sequences, intronic or non-coding variants, mitochondrial DNA, epigenic variants or balanced rearrangements [43].

A recent metanalysis reviewed the diagnostic yield of ES in NDDs and found an overall diagnostic yield of 36% [28], which is well above that of CMA. The yield was 54% in syndromic NDDs and 31% in isolated NDDs [28]. While several studies examined the impact of various clinical features on the diagnostic yield of ES, none reported any statistically significant differences [44,45,46]. Higher yields were observed with abnormal head size (microcephaly and macrocephaly), developmental epileptic encephalopathy and a younger age at presentation [44,47,48,49,50]. In many studies, the yield was equivalent in syndromic and non-syndromic ID [44,45,46]. Periodic re-analysis of ES (every 1–3 years) can enhance diagnostic yield over time by 10–16% [51,52,53].

More recently, small exon-level insertion/deletion (Indel) calling is being incorporated into ES bioinformatic pipelines and has been shown to further increase diagnostic yield. In a recent study, exome-based single nucleotide variant (SNV) and Indel calling combined with exome-based CNV analysis in ES data from patients with NDD, revealed an overall diagnostic yield of 54.0% (35.1% from SNV/Indel and 18.9% from exome based CNV) [54]. A similar study explored diagnostic yield in unexplained DD/ID using exome-based exon-level Indel and CNV analysis, and reported an overall diagnostic yield of 58.8% (41.2% from SNV/Indel constitute and 17.6% CNV) [55].

Comprehensive NGS-based GDD/ID gene panels simultaneously sequence multiple genes (usually over 2000) associated with GDD/ID. A few studies have explored the diagnostic yield of comprehensive GDD/ID gene panels with reported figures of 11–39% [56,57,58,59]. Studies that compared “simulated” panels to ES demonstrated slightly lower diagnostic yields in panels vs. ES [45,60]. Multi-gene panels are usually performed on an exome back-bone, where variants are reported in only selected genes from the exome. Analysis of the remainder of the ES data is at times possible, depending on the providing clinical laboratory.

Trio testing refers to the testing of the proband and both biological parents to help identify and interpret suspected gene variants in the proband. Several studies have reported higher yield from trio testing than from proband-only testing [28,49,61,62,63,64]. Furthermore, additional advantages of trio-based testing include decreased resources for analysis, variant testing in parents and earlier time to definitive diagnosis.

Many of the professional practice guidelines do not currently formally recommend ES testing in GDD/ID, as many were published before ES and NGS testing was readily available in clinical practice. However, ES and comprehensive gene panels are routinely used as an integral part of the genetic evaluation of individuals with GDD/ID. The recent 2021 ACMG guidelines strongly recommend ES as first or second line testing in individuals with GDD/ID [21,28].

Therefore, ES, or comprehensive GDD/ID gene panels, should be considered in the specialty evaluation of children with GDD/ID, given their high yield (above all other testing). Early use of ES in the diagnostic journey, when possible, is recommended. Use will likely be dependent on locally available supports (including financial) and testing resources.

### 2.3. Genome Sequencing

Genome sequencing (GS) is an NGS approach that determines the sequence of most of the DNA of the entire genome of an individual. The main advantage of GS over ES is the ability to query intronic regions, and its superior ability compared with ES to detect structural rearrangements including small and large CNVs [43,65]. GS is not yet clinically available in most centers, and its use is still mainly restricted to research paradigms. One study demonstrated that the addition of GS to the investigation of patients with GDD/ID following unrevealing initial testing (either CMA, ES or both) had a diagnostic yield of 21% [65]. The yield was 64% if only a CMA had been previously performed and 14% if ES was performed [65].

GS is not yet recommended by any professional organization given its limited availability. However, it is likely that GS will supplant ES and CMV in the foreseeable future as decreasing testing costs and increased accessibility to clinicians emerge.

### 2.4. Fragile X Syndrome Testing

Fragile X syndrome (FXS) is an X-linked triplet repeat expansion disorder caused by the unstable expansion of CGG repeats in the 5′untranslated region of the *FMR1* gene. FXS is one of the most common monogenic causes of GDD/ID with a prevalence of 1.4:10,000 males and 0.9:10,000 females. Though it is characterized by distinctive clinical features, including distinctive facial dysmorphisms (long face, large ears, prominent jaw) and macro-orchidism, these may be subtle or only apparent with entry into puberty. Fragile X testing in a cohort of 2486 individuals with NDD demonstrated a yield of 1.2% [66]. Furthermore, 96% of the FXS-positive cases had clinical features of FXS or a positive family history [66]. The yield has been shown to be significantly higher when testing is restricted to males with NDD with characteristic physical/behavioral features or family history, at 9.5–17% [67,68,69].

FXS testing is widely available. It is important to note that FXS cannot be diagnosed by CMA, ES or gene panels. Given the fact that it is an X-linked condition, prompt diagnosis is critical as it will have important impact on recurrence risk and genetic counseling.

Most professional organizations presently recommend testing for FXS as first line in all individuals with GDD/ID.

### 2.5. Metabolic/Biochemical Screening for Inherited Metabolic Diseases

Inherited metabolic disorders (IMDs) are genetic disorders that result in metabolic defects due to a deficiency of enzymes, membrane transporters or other functional proteins. Consideration of IMDs when evaluating individuals with GDD/ID is important, especially when early detection can lead to specific treatments that improve clinical outcomes. Over 100 IMDs associated with GDD/ID have potential treatment [26,27]. Furthermore, since these conditions are largely autosomal recessive, diagnosis has as impact on genetic counseling, as risk of recurrence in subsequent pregnancies is often elevated in a Mendelian fashion.

Biochemical screening should be performed in any individual displaying red flags suggestive of an IMD. These include, but are not limited to, developmental plateau or regression in the context of an abnormal exam, altered level of consciousness, observed movement disorders (such as chorea, dystonia, ataxia, and myoclonus), hepatomegaly, splenomegaly, drug-resistant seizures, coarse facial features and multisystemic involvement crossing embryonic origin of affected tissues. Individuals with these features should also be urgently referred to biochemical genetic specialists. Screening metabolic testing may include serum ammonia, lactate, copper, ceruloplasmin, homocysteine, plasma amino acids, urine organic acids, purines, pyrimidines, creatine metabolites, oligosaccharides, and glycosaminoglycan.

There is no current consensus regarding whether biochemical screening for treatable IMDs should be performed in children with non-syndromic GDD/ID in the absence of red flags. Both the Canadian Pediatric Society and the American Academy of Pediatrics recommend the systematic biochemical screening in all children with GDD/ID [10,24,26], while others recommended screening only if clinical features were suggestive of IMDs [23,70,71].

The diagnostic yield of metabolic screening in non-syndromic GDD/ID is extremely low, estimated at 0.25%−0.42% in a non-consanguineous population [71,72]. A review of the implementation of Treatable Intellectual Disability Endeavor (TIDE) screen protocol for cases of GDD/ID without clinical features suggestive of IMDs showed no significant increase in the diagnosis of IMDs, despite a four-fold increase in testing [71].

Given the low yield, routine screening for IMDs is not presently recommended without the presence of clinical features suggestive of IMDs, though should be considered if newborn screening was not previously performed.

## 3. Common CNVs in Non-Syndromic GDD/ID

Several microdeletions and duplications have been associated with a wide range of phenotypic features in patients with both syndromic and non-syndromic GDD/ID. A list of common CNVs associated with non-syndromic GDD/ID is provided in Table 1.

In addition, several of the microdeletion syndromes that are associated with syndromic GDD/ID, such as Prader–Willi and Angelman syndromes (deletion of 15q11-q13), Williams–Beuren (deletion of 7q11.23) and Smith–Magenis syndrome (deletion of 17p12) [73], may present milder phenotypes with unrecognizable syndromic features. For example, in a recent study that identified pathogenic CNVs in 33 patients with non-syndromic GDD/ID, one individual was found with 15q11.2 deletion but absent characteristic features of Angelman/Prader–Willi syndromes. The same study also showed other frequent aneusomies in chromosomes 15, 16 and X [41].

**Table 1 children-10-00414-t001:** Common microdeletions and microduplications associated with non-syndromic GDD/ID.

Chromosome Region	Deletion or Duplication	Main Clinical Features	Candidate Genes	References
15q11.2	Deletion/Duplication	ID, schizophrenia, epilepsy	*TUBGCP5, CYFIP1, NIPA2, NIPA1*	[74,75,76,77,78,79,80]
15q13.3	Deletion	ID, epilepsy, schizophrenia, ASD	*CHRNA7*	[74,81,82,83,84,85,86]
16p11.2 (distal)	Deletion	ID, ASD, obesity, schizophrenia	*SH2B1*	[87,88]
16p11.2 (proximal)	Deletion/Duplication	ID, language delay, ASD, obesity	*MVP, CDIPT1, SEZ6L2, ASPHD1, KCTD13*	[89,90,91,92,93,94,95,96]
16p12	Deletion	Intellectual disability	*UQCRC2, EEF2K, POLR3E, CDR2*	[41,97]
Xq28	Duplication	Males: hypotonia, severe GDD and ID, progressive spasticity, seizures, ASD Females: milder phenotype	*RAB39B, CLIC2, IRAK1, MECP2, GDI1*	[41,98,99,100,101]

## 4. Common Pathways Underlying NDDs

With the current expanding number of genes associated with NDDs, one interesting approach toward a better understanding of their complex genetic heterogeneity is to understand the common pathways in which GDD/ID genes are acting. Since patients with non-syndromic GDD/ID present intellectual impairment as their main clinical feature, it is logical that causative genes function in processes related to learning and memory. Key cellular mechanisms involved in GDD/ID pathogenesis include those related to synaptic structure and signaling, protein homeostasis and epigenetic regulation of transcription [102]. Furthermore, altered neural circuits in multiple brain areas, including the cerebral cortex, basal ganglia and thalamus, have been described in ID/GDD [103]. A summary of these pathways is provided in Table 2. Figure 1 illustrates the interplay between these pathways.

### 4.1. Synaptic Signaling Dysregulation

Many genes associated with GDD/ID encode molecules with important roles in the assembly, structure, and function of the neuronal synapse.

Synaptic cell-surface proteins, such as the neurexins (NRXN) and neuroligins (NLGN), have important roles in the assembly of functional synapses, and post-synaptic scaffolding proteins, such as members of the SHANK family, have an important role in organizing the architecture and molecular composition of synapses [102]. Their encoding genes, *NRX1*, *NLGN3*, *NLGN4*, *SHANK2* and *SHANK3*, respectively, have been previously associated with a wide range of NDDs, including GDD, ID and autism spectrum disorder (ASD) [104,105,106].

Impaired glutamate excitatory signaling has also been associated with NDDs. The two main ionotropic glutamatergic channels found in the post-synaptic density are α-amino-3-hydroxy-5-methyl-4-isoxazolepropionic acid (AMPA) and N-methyl D-Aspartate (NMDA) receptors. Mutations in genes encoding components of glutamatergic receptors such as *GRIK2*, *GRIN2B* and *GRIA2* have been associated with non-syndromic ID and other neurodevelopmental abnormalities, including ASD [107,108,109].

The brain-specific synaptic Ras/Rap GTPase-activating protein, encoded by *SYNGAP1*, is another critical component of the postsynaptic density [102]. It suppresses the signaling of pathways linked to the NMDA receptor, such as the Ras/ERK pathway [110,111,112]. De novo heterozygous loss-of-function mutations in *SYNGAP1* are associated with non-syndromic ID, ASD and epilepsy [113].

The membrane-associated guanylate kinase (MAGUK) proteins are another group of multidomain scaffolding proteins in the postsynaptic density. Impaired synapse formation due to incorrect localization of the MAGUK family protein PSD-95 (DLG4) was reported in association with mutations in the interleukin 1 receptor accessory protein-like 1 (*IL1RAPL1*) [3]. Pathogenic variants in this gene were also previously associated with non-syndromic ID and ASD [114]. Mutations in another MAGUK family protein gene, *CASK*, which encodes a calcium/calmodulin-dependent serine protein kinase, were also previously reported in patients with non-syndromic ID [3,115].

NDDs have also been associated with disorders in neurotransmitter release at the pre-synaptic membrane. The SNARE complex is responsible for mediating the fusion of the vesicle with the pre-synaptic membrane and pathogenic variants in *STXBP1*, one of its subunits, are classically associated with severe infantile epileptic encephalopathy [110,116]. However, variants in this gene have also been reported in individuals with isolated ID [117].

### 4.2. Protein Homeostasis

Protein homeostasis is a key regulatory process of synaptic plasticity and assembly in cortical and basal ganglia circuits [103].

The phosphatidylinositol 3-kinase (PI3K)-mTOR pathway is major signaling cascade implicated with the emerge of well-known syndromic ID and ASD-related syndromes, such as tuberous sclerosis, Cowden and Fragile X [110].

The ERK/MAPK is another important signaling cascade required for synaptic plasticity [3]. Besides the previously mentioned SynGAP protein that negatively regulates the pathway, mutations in the *RPS6KA3* gene, a member of the ribosomal S6 kinase family that functions as a downstream effector of the ERK signaling pathway, are also related to non-syndromic ID [118].

Other findings highlight that deficient protein synthesis is a common mechanism in non-syndromic NDDs. De novo loss-of-function mutations in *CUL3,* which encodes a component of the Cullin-RING ligase (CRL) complex that controls proteasomal degradation of specific target proteins, have been associated with non-syndromic ASD [119,120]. In mouse models, protein translation deficiencies have also been linked to autism-associated variants in the synaptic adhesion molecule neuroligin-3 (*NLGN3*). The loss of *NLGN3* was accompanied by a disruption of homeostasis in the ventral tegmental area [121].

### 4.3. Epigenetic Regulation

Increasing evidence supports the dysregulation of transcriptional and epigenetic control of gene expression as mechanisms underlying both syndromic and non-syndromic NDDs [103]. DNA methylation and histone post-translational modification (e.g., acetylation, methylation and phosphorylation) are the main molecular changes implicated in the epigenetic regulation of transcription [110].

*MECP2* encodes methyl CpG-binding protein 2 and is believed to act as a transcriptional modulator by binding methylated CpG DNA [110]. Although traditionally associated with Rett syndrome, mutations in *MECP2* are also described in sporadic cases of autism [122,123]. Mutations in chromatin regulators, such as the chromodomain helicase DNA-binding protein 8 (*CHD8*) gene, have also been associated with ID and ASD [124].

### 4.4. Thalamic and Peripheral Circuits

Altered tactile sensitivity is a common feature in individuals with GDD/ID and autistic features. Thalamic circuits are responsible for processing sensory signaling from the periphery to the cerebral cortex. Following that idea, autistic symptoms found in individuals with GDD/ID are now being related to thalamic dysfunction [103].

Loss of function mutations in the patched domain containing protein 1 (*PTCHD1*) were previously associated with an X-linked neurodevelopmental disorder with a strong propensity to autistic behaviors [125], suggesting that this protein has a significant role in cognitive and attentional control. Interestingly, enriched expression of the *PTCHD1* in the thalamic reticular nucleus of mice was associated with inhibitory inputs to the thalamus [103]. In addition, another study showed that mice with ASD-linked genes (including previously mentioned *MECP2* and *SHANK3*) deleted selectively from mechanosensory neurons presented altered tactile sensitivity and discrimination [126]. *SHANK3* and *MECP2* variants have also been shown to be responsible for alterations in basal ganglia circuits in animal models [127,128,129,130].

## 5. Diagnostic Approach to the Evaluation of Children with Non-Syndromic GDD/ID

An overview of the evaluation of individuals with non-syndromic GDD/ID is presented in Table 3.

Careful history taking and thorough clinical examination is the first step in the evaluation of children with GDD/ID, with particular attention to the identification of an acquired etiology and recognition of clinical features that could be suggestive of a syndromic cause. A detailed three-generation family history, prenatal and birth history as well as neurodevelopmental histories should be obtained. If a clear acquired cause is identified, further genetic testing is not warranted. If a known syndrome is recognized based on the clinical features (e.g., Rett syndrome or Zellweger syndrome), then targeted genetic or metabolic testing should be ordered first. It is important to highlight that individuals with an underlying genetic disorder are at a higher risk of also having an additional acquired neurologic insult; for example, children with neuromuscular disorders and hypotonia are at a higher risk of neonatal asphyxia. Therefore, clinicians should keep in mind that both acquired and genetic etiologies may co-occur in some patients.

Formal visual and hearing assessment should be performed in all children with GDD/ID due to the high frequency of primary sensory impairments and their potential for amelioration and remediation [10,131,132,133].

Brain imaging should be performed in the presence of abnormal head circumference, rapid change in head circumference over time, focal neurological findings or focal epilepsy [134]. Brain imaging is more likely to detect abnormalities when MRI is performed in individuals with GDD/ID with additional clinical or neurological signs [135,136]. In one study, abnormal imaging findings were observed in 41% of individuals when selective imaging was performed compared with a yield of 14% with non-selective screening [137]. MRI brain is generally preferred when compared with CT brain given its higher sensitivity [23,24,132]. Most professional organizations recommend brain imaging especially when other neurological findings are present [24]. The addition of MR Spectroscopy can be considered to aid in the diagnosis of suspected mitochondrial disorders or cerebral creatine deficiency syndrome.

Screening for IMDs is not presently recommended for non-syndromic GDD/ID in the absence of red flags. Metabolic screening should be considered in children who have not undergone newborn metabolic screening, as performed in many countries.

Genetic testing should be performed in all children with GDD/ID in whom an initial evaluation did not uncover an etiology. All individuals should undergo CMA testing and ES. When ES is not available, a comprehensive GDD/ID gene panel is a good substitute. Trio testing is preferred given easier analysis and no need to obtain additional samples for variant segregation. FXS testing also should be performed in all patients. CMA and FXS testing are generally recommended as first-tier as they are easily accessible, less costly and can be ordered by primary providers. ES, given its higher cost and more limited accessibility to non-specialists and non-geneticists, is usually ordered as a second-tier test upon specialty evaluation.

## 6. Overview of Management Principles for Children with GDD/ID

The comprehensive care of individuals with GDD/ID is based on a family-centered, multidisciplinary approach. Multiple medical expertises and different subspecialists, including pediatricians, pediatric neurologists, geneticists, psychiatrists, family practitioners, physiatrists and orthopedic surgeons, amongst others, are often necessary to provide ideal care [138]. Furthermore, the involvement of occupational therapists, physiotherapists, psychologists, special educators, nurses and social workers is fundamental, and enables the minimization of impairments and possible complications, and the maximization of the activity, participation, health and well-being of patients [139]. The challenges and needs of the patients evolve over their lifespan [140].

Medical issues that are relatively frequently encountered in children with GDD/ID include epilepsy, behavioral challenges, sleep disturbances, movement disorders and orthopedic deformities [140].

General principles for antiepileptic drug treatment include selection of the drug based on seizure type, avoidance of polypharmacy and minimal use of sedating or cognitively depressing antiepileptic drugs, as well as those that can exacerbate pre-existing behavioral challenges (i.e., Levetiracetam) [110].

Behavioral challenges possibly represent the most difficult issue for caregivers. Psychologic/psychiatric intervention is frequently the first line of approach recommended, but pharmacologic treatment can be necessary for handling disorders of attention and impulsivity (with stimulant or non-stimulant alternatives), agitation, opposition, disinhibition and aggression towards both others and self, in which cases atypical antipsychotics have shown a good response [110].

Sleep disturbances are highly prevalent in children with neurodevelopmental disabilities, and significantly impact the children and the family’s quality of life. Proper sleep hygiene is a first step, followed by the use of melatonin, and, if needed, other medications such as gabapentin, clonidine and trazodone [141].

Movement disorders may include prominent spasticity with possible progression to orthopedic deformities. Physiotherapy is often a good preventive measure, but in those cases with refractory pain or functional limitations intramuscular botulin toxin injections and oral antispasmodic agents could be helpful.

## 7. Conclusions

While detailed history and clinical examination remains a cornerstone in the evaluation of children with GDD/ID, genetic testing is essential and key in individuals in whom the etiology is not readily apparent. In the era of genomic first approach, the underlying genetic causes of non-syndromic GDD/ID can be explored with a wide variety of diagnostic assessments and result in high etiologic yields, allowing a specific genetic diagnosis. As technologies progress and evidence emerges, approaches regarding genetic testing will evolve over time. Reaching a specific genetic diagnosis provides many advantages to patients and families, including an end of the diagnostic odyssey, precise genetic counseling and optimized longitudinal management and targeted surveillance. Though targeted treatments and gene therapies are limited in GDD/ID, advances in this area are intensively being investigated and their application in the clinical setting is anxiously awaited.

## Figures and Tables

**Figure 1 children-10-00414-f001:**
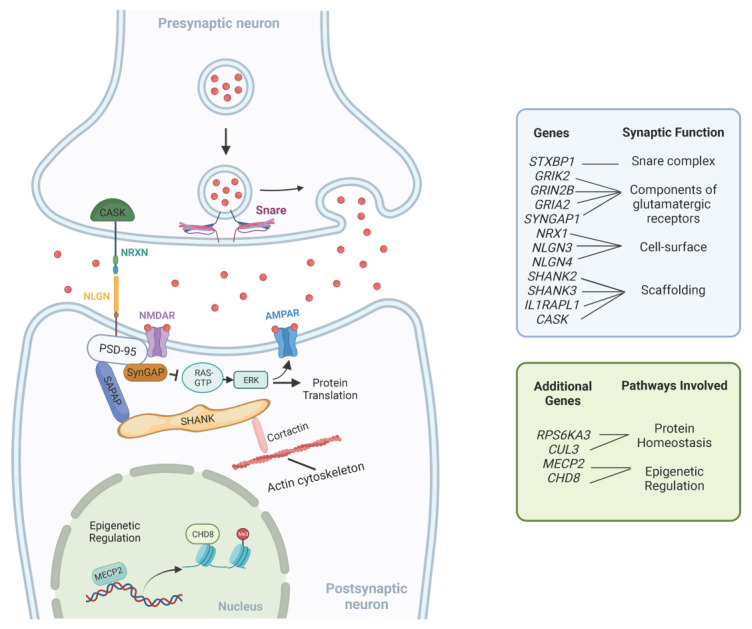
Representation of common pathways underlying GDD/ID. This figure is not exhaustive and only shows a few examples of molecules involved in synaptic function, epigenetic regulation and protein homeostasis. Adapted from “Tripartite Glutamatergic Synapse”, by BioRender.com (2023). Retrieved from https://app.biorender.com/biorender-templates. Accessed on 10 February 2023.

**Table 2 children-10-00414-t002:** Summary of common pathways and examples of genes associated with non-syndromic GDD/ID and their respective functions.

Common Pathways	Genes	Function
**Synaptic** **Signaling**	*NRX1*	Cell-surface receptors that bind neuroligins; required for efficient neurotransmission.
	*NLGN3 NLGLN4*	Mediate cell-to-cell interactions between neurons; linked to glutamatergic postsynaptic proteins.
	*SHANK2 SHANK3*	Scaffolding and cell adhesion proteins; required for synaptic plasticity.
	*GRI2K GRIN2B GRIA2*	Subunits of synaptic glutamate receptors; required for neurotransmission.
	*SYNGAP1*	Part of the NMDA receptor complex; involved in negative regulation of ERK/MAPK pathway.
	*IL1RAPL1*	Part of the interleukin 1 receptor; required for neuronal calcium-regulated vesicle release and dendrite differentiation.
	*CASK*	Part of the MAGUK family; scaffolding proteins.
	*STXBP1*	Synaptic vesicle docking and fusion; required for efficient neurotransmission.
**Protein** **Homeostasis**	*RPS6KA3*	Part of the RSK (ribosomal S6 kinase) family of growth-factor-regulated serine/threonine kinases; involved in ERK/MAPK pathway.
	*CUL3*	Part of the ubiquitin-proteasome system; required for proteasomal degradation of unwanted proteins.
**Epigenetic** **Regulation**	*MECP2*	Chromatin-associated protein involved in methyl binding to control transcription; required for maturation of neurons.
	*CHD8*	ATP-dependent chromatin-remodeling factor that regulates transcription.

**Table 3 children-10-00414-t003:** Approach to evaluation of children with non-syndromic GDD/ID.

Evaluation	Recommendation
1. Detailed history including developmental and family history with thorough clinical examination	▪ If consistent with acquired cause → No genetic testing required. ▪ If specific genetic etiology is suspected → Proceed with appropriate targeted genetic/metabolic test. ▪ If abnormal head circumference, focal neurological finding, or epilepsy present → Consider MRI brain. ▪ If no prior newborn screen or clinical signs suggestive of IMDs → Consider TIDE protocol. ▪ If any red flags suggestive of IMD → Urgent referral to metabolic/biochemical specialist. ▪ Do not forget to perform hearing and visual evaluation. ▪ If no specific etiology identified: Proceed to next step.
2. First tier genetic testing	▪ Start with CMA and FXS testing. ▪ If possible, ES or Multigene GDD/ID Panel (Trio is preferred). ▪ If first tier testing is negative → Refer to genetic subspecialist for second tier genetic testing.
3. Second tier genetic testing	▪ Send for ES or Multigene GDD/ID panel if not already done. ▪ Negative ES → Consider reanalysis in ES every 1–3 years. ▪ Negative Multigene panels with ES backbone → Proceed with ES. ▪ If still negative proceed to next step.
4. Further investigations	▪ Consider GS if available.

ES: exome sequencing, FXS: Fragile X syndrome, GDD: global developmental delay, GS: genome sequencing, IMD: inherited metabolic disorders, TIDE: Treatable Intellectual Disability Endeavor.

## Data Availability

Not applicable.

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
