# Peer review of "Evaluation of Individuals with Non-Syndromic Global Developmental Delay and Intellectual Disability"

_children, 2023, doi:10.3390/children10030414_

Round 1

Reviewer 1 Report

No comments 

Author Response

We thank the reviewer for their positive evaluation. There are no specific comments that we need to respond to.

Reviewer 2 Report

This paper provides a highly readable and accurate summary of how and when to consider ordering genetic testing for children with unexplained, non-syndromic GDD/ID. The paper also provides well-sourced recommendations for neuroimaging and metabolic testing of such children,  though not promised by the title, so I would suggest a revision to the title to, say, "Diagnostic evaluation..." rather than just "Genetic evaluation..." and perhaps mentioning the other testing in the abstract.

Author Response

We thank the reviewer for their positive comments.

As suggested, we have changed the title of our review from "Genetic evaluation of individuals with non-syndromic global developmental delay and intellectual disability", to the more general "Evaluation of individuals  with non-syndromic global developmental delay and intellectual disability" as we discuss non-genetic testing (metabolic and imaging) as well as a general approach to management. 

Reviewer 3 Report

The authors summarized common procedures and further tests for children with potential GDD or ID and proposed a protocol for future evaluation, which could be useful for both pediatricians and patient families. 

On line 244, there is a redundant period. The authors should double-check their manuscript to correct any typos or mistakes.

There is one comment to the authors about the standards for GDD and ID. Given that environmental toxins have increased these years, do the authors notice any growth delay for the average child compared to years before? Are there increased cases of GDD or ID, or have the standards changed due to overall growth delay? If that's the case, is it necessary to increase testing, as maybe more children are potentially affected?

Author Response

We thank the reviewer for their valuable comments.

As suggested, we have updated the paper and have included a brief discussion on the prevalence of ID/GDD over years and across different levels of socioeconomic status, and included that the postulated increased prevalence in developing countries and regions with low socio-demographic index is due to higher exposure to risk factors and environmental influences

" The prevalence of ID is variable across different global populations, spectrum of severity and socioeconomic levels [3–5]. A recent study, the global prevalence of ID was found to have decreased from 1.74% in 1990 to 1.39% in 2019 [5]. Given the overall increase of the global population, the total number of individuals with ID was larger overtime [5]. There is a higher proportion of individuals with ID in the regions of low-middle socio-demographic index compared to high socio-demographic index (5-6 times higher) [5]. The higher prevalence of ID in developing countries and low socioeconomic areas was postulated to be due to higher exposure to risk factors and environmental influences [5]."

We have not addressed growth delay as this is out of the scope of this review.